# Deep Reinforcement Learning for Charging Scheduling of Electric Vehicles Considering Distribution Network Voltage Stability

**DOI:** 10.3390/s23031618

**Published:** 2023-02-02

**Authors:** Ding Liu, Peng Zeng, Shijie Cui, Chunhe Song

**Affiliations:** 1Key Laboratory of Networked Control Systems, Shenyang Institute of Automation, Chinese Academy of Sciences, Shenyang 110016, China; 2Institutes for Robotics and Intelligent Manufacturing, Chinese Academy of Sciences, Shenyang 110016, China; 3Shenyang Institute of Automation, Chinese Academy of Sciences, Shenyang 110016, China; 4University of Chinese Academy of Sciences, Beijing 100049, China

**Keywords:** electric vehicle, distribution network, deep reinforcement learning, voltage control

## Abstract

The rapid development of electric vehicle (EV) technology and the consequent charging demand have brought challenges to the stable operation of distribution networks (DNs). The problem of the collaborative optimization of the charging scheduling of EVs and voltage control of the DN is intractable because the uncertainties of both EVs and the DN need to be considered. In this paper, we propose a deep reinforcement learning (DRL) approach to coordinate EV charging scheduling and distribution network voltage control. The DRL-based strategy contains two layers, the upper layer aims to reduce the operating costs of power generation of distributed generators and power consumption of EVs, and the lower layer controls the Volt/Var devices to maintain the voltage stability of the distribution network. We model the coordinate EV charging scheduling and voltage control problem in the distribution network as a Markov decision process (MDP). The model considers uncertainties of charging process caused by the charging behavior of EV users, as well as the uncertainty of uncontrollable load, system dynamic electricity price and renewable energy generation. Since the model has a dynamic state space and mixed action outputs, a framework of deep deterministic policy gradient (DDPG) is adopted to train the two-layer agent and the policy network is designed to output discrete and continuous control actions. Simulation and numerical results on the IEEE-33 bus test system demonstrate the effectiveness of the proposed method in collaborative EV charging scheduling and distribution network voltage stabilization.

## 1. Introduction

In recent years, electric vehicle (EV) technology has developed rapidly, driven by breakthroughs in battery technology [1,2]. As a substitute for fossil fuel vehicles, EVs have received extensive attention due to their environmentally friendly characteristics [2,3,4,5]. EVs can reduce traffic pollution emissions and have lower charging costs than refueling, which has been widely accepted and deployed [6,7,8,9]. However, large-scale numbers of EVs connected to the power grid will bring challenges, such as frequency excursion and voltage fluctuation [10]. The voltage stability of the distribution network (DN) is an issue that needs to be focused on. The uncontrolled charging process of EVs will affect the voltage stability of the distribution network [11]. When EVs are connected to the power grid in vehicle-to-grid (V2G) mode, this situation will further deteriorate. In addition, the access of distributed generators (DGs) to the power system changes the direction of power flow, and the injection of active power upstream by distributed generators (DGs) causes voltage rise and interferes with Volt/Var control (VVC) equipment [12]. The intermittency, randomness, and fluctuation of renewable energy sources (RESs) can cause voltage fluctuations in the distribution network [13].

VVC is used to improve the voltage stability of the distribution network. In traditional VVC practice, voltage-regulating devices, such as on-load tap changers (OLTCs), voltage regulators (VRs), and switchable capacitor banks (SCBs), are leveraged to mitigate voltage violations [14]. In [15], a support vector regression based model predictive control (MPC) method was proposed to optimize the voltage of a distribution network. For the renewable energy access problem controlled by inverters, a multistage method has been widely used for the VVC of distribution networks [16,17]. To coordinate VVC equipment and distributed power supply, neural networks and learning-based methods are widely used. An artificial neural network (ANN) based approach was introduced for the VVC of distributed energy resources at the grid edge [18]. The authors of [19] proposed a safe off-policy deep reinforcement learning algorithm for VVC in a power distribution system. A model-free approach based on constrained safe deep reinforcement learning was proposed in [12] to solve the problem of optimal operation of distribution networks. Although the aforementioned research has made some achievements in VVC, they did not consider the effect of electric vehicle charging on voltage stability nor did they consider the possibility of electric vehicles participating in voltage control.

EVs participate in the voltage regulation of distribution networks. On the one hand, the charging behavior of EVs is stimulated by the electricity price; on the other hand, EVs can operate in the V2G mode [20]. By adjusting the charging power or discharge power of EVs, it is helpful to stabilize the voltage of the distribution network [21,22,23,24]. Researchers have performed much work on the problem of charging scheduling of electric vehicles. The authors of [25] proposed an improved binary particle swarm optimization (PSO) approach to solve the problem of the controlled charging of EV with the objective of reducing the charging cost for EV users and reducing the pressure of peak power on the distribution network. To avoid the limitations of deterministic methods in terms of models and parameters and their inability to handle real-time uncertainty, deep reinforcement learning is widely used in the charging scheduling problem for EVs. References [3,10] proposed model-free approaches based on deep reinforcement learning and safe deep reinforcement learning, respectively, for the charging scheduling of household electric vehicles. Both consider the uncertainty of the system and do not need an accurate model but only study the charging scheduling problem of home electric vehicles. When faced with the problem of charging EVs on a larger scale, the charging process for EVs is managed by an aggregator or central controller. However, the charging process of EVs is highly uncertain, which requires the estimation and prediction of the charging demand of EVs. Artificial intelligence approaches are currently of interest due to their advantages in dealing with high-dimensional data and non-linear problems. A Q-learning-based prediction method was proposed in [26] for forecasting the charging load of electric vehicles under different charging scenarios. The authors of [27] proposed a demand modeling approach based on 3D convolutional generative adversarial networks. Reference [28] designed a deep learning-based forecasting and classification network to study the long-term and short-term characteristics of the charging behaviors of plug-in EVs. To solve the problem of EV cluster charging, [29] proposed a hybrid approach to reduce the power loss and improve the voltage profile in the distribution system, and both the vehicle-to-grid and grid-to-vehicle operational modes of EVs were considered in this work. However, the above research only studies the charging problem of electric vehicles from the perspective of demand response (DR). The capacity and access location of the EV charging load will affect the power flow distribution of the distribution network, and disordered electric vehicle charging will reduce the voltage stability of the distribution network. The authors of [30] proposed an evolutionary curriculum learning (ECL)-based multiagent deep reinforcement learning (MADRL) approach for optimizing transformer loss of life while considering various charging demands of different EV owners. This work only focuses on the life of the transformer and does not directly control the voltage. Reference [20] proposed a three-layer hierarchical voltage control strategy for distribution networks considering the customized charging navigation of EVs. Although the hourly scheduling results of the OLTC are given the day before, the voltage is controlled in minutes, and frequent voltage regulation will reduce the life of the OLTC.

The above analysis shows that the current research is more concerned with the VVC of DN or DR of EVs, and there are fewer studies that consider both and perform coordinated optimization. However, the studies that do examine the coordinated optimization of both do not consider the actual system comprehensively. The collaborative optimization of EVs, schedulable DGs and VVC devices in an DN system faces some challenges. First, the charging goals of EV users and the goal of maintaining voltage stability in the distribution networks are mutually exclusive. Second, the distribution network has strong uncertainty and nonlinearity, and the charging process of EVs has strong uncertainty due to arrival time, departure time, and electricity price. Third, there are many homogeneous devices controlled by discrete and continuous actions in the system.

To solve these challenges, we formulate a collaborative EV charging scheduling and voltage control strategy based on DRL to comprehensively schedule the charging of EVs and control the voltage of distribution networks. We establish an MDP model for the charging scheduling of EVs and the voltage control problems of distribution networks. The state variables of the system take into account the uncertainty of the EV charging process, nodal loads, RES generation, and electricity price interacting with the main grid. The purpose is to realize automatic voltage regulation and reduced EV charging cost though the collaborative control of VVC devices and EVs, as well as controllable DGs. The design of the reward function comprehensively considers the charging target of EVs and the voltage control objective of DN. In contrast to the control strategies mentioned in the literature above, which were graded according to time, the proposed control strategy synergistically considers the problem of optimizing the scheduling of EVs and the voltage control of the DN. The collaborative scheduling control strategy consists of two layers; the upper layer manages the charging of electric vehicles and the lower layer regulates the voltage control equipment. The control strategy is output by a designed deep neural network (DNN) and trained using a model-free deep deterministic policy gradient (DDPG) method. A signal rounding block is set up after the output layer of the DNN to obtain the discrete control signals of VVC devices. The main contributions of this work are:A time-independent two-layer coordinated EV charging and voltage control framework is proposed to minimize EV charging costs and stabilize the voltage of distribution networks.An MDP with unknown transition probability is established to solve the EV charging problem considering the voltage stabilization of DN. The reward function is reasonably designed to balance the EV charging target and voltage stability target.The model-free DDPG algorithm is introduced to solve the coordinated optimization problem. A DNN-based policy network is designed to output hybrid continuous scheduling signals and discrete control signals.

The rest of the paper is organized as follows: Section 2 presents the MDP model and introduces the collaborative scheduling control strategy. Section 3 encompasses the simulation experiments and analysis. Section 4 gives the conclusions.

## 2. Materials and Methods

### 2.1. Modelling of DN System

In this paper, we propose a collaborative EV charging and voltage control framework on a DN system. As shown in Figure 1, EVs, controllable DGs, and RES are distributed in the DN system. EVs are controlled by smart terminals that control the charging process and pay for charging. The central controller (CC) collects information on the operating status of the system though two-way real-time communication and based on this information, outputs signal to control the controllable units through our proposed collaborative EV charging and voltage control strategy. In the formula we established, the total operating time range of DN is divided into T time slots and the subscript t represents the specific time slot. In the DN system, the subscript $i\in \Omega_{n}$ is used to represent the nodes and $\Omega_{n}$ is the set of all nodes, the subscript $ij\in \Omega_{b}$ is used to represent the branches, and $\Omega_{b}$ is the set of all branches. We then perform the detailed modelling of EVs, controllable DGs, and voltage control devices in the DN system. The operational constraints of DN are subsequently given, and the MDP model is finally established. It is worth noting that our model does not require knowledge of the topology of the DN, the specific line parameters, and the distribution and fluctuations of the load. The scheduling strategy is learned only according to the observed system state.

#### 2.1.1. Controllable Units in the Distribution Network

1.EVs

The control variable of an EV is its charging power and, in V2G mode, discharge power. For EVs connected to node i, the control variable is expressed as $P_{i,t}^{\mathrm{EV}}$, and in V2G mode, the value of $P_{i,t}^{\mathrm{EV}}$ is positive for charging and negative for discharging. Under the intelligent charging strategy, the charging power constraints are:(1)$$-P_{i,\mathrm{dis},\max}^{\mathrm{EV}}\leq P_{i,t}^{\mathrm{EV}}\leq P_{i,\mathrm{ch},\max}^{\mathrm{EV}},i\in \Omega_{n},t\in \Omega_{t}$$
where $P_{i,\mathrm{ch},\max}^{\mathrm{EV}}$ and $P_{i,\mathrm{dis},\max}^{\mathrm{EV}}$ are the maximum charging power and maximum discharging power of EV.

The SOC represents the state of charge of the battery, which should meet the following constraints during any scheduling period t [31]:(2)$$SoC_{i,t+1}^{\mathrm{EV}}=\left\{\begin{array}{c} \begin{array}{cc} SoC_{i,t}^{\mathrm{EV}}+P_{i,t}^{\mathrm{EV}}\cdot \eta_{i,\mathrm{ch}}^{\mathrm{EV}}\cdot \Delta t/E_{i}^{\mathrm{EV}},\;\; & \;\;if\;P_{i,t}^{\mathrm{EV}}\geq 0 \end{array}\\ \begin{array}{cc} SoC_{i,t}^{\mathrm{EV}}+P_{i,t}^{\mathrm{EV}}\cdot \Delta t/\left(\eta_{i,\mathrm{dis}}^{\mathrm{EV}}\cdot E_{i}^{\mathrm{EV}}\right), & if\;P_{i,t}^{\mathrm{EV}}<0 \end{array} \end{array}\right.,\;i\in \Omega_{n},t\in \Omega_{t}$$
(3)$$SoC_{i,\min}^{\mathrm{EV}}\leq SoC_{i,t}^{\mathrm{EV}}\leq SoC_{i,\max}^{\mathrm{EV}},i\in \Omega_{n},t\in \Omega_{t}$$
where $E_{i}^{\mathrm{EV}}$ is the capacity of the EV battery, $\eta_{i}^{\mathrm{EV}}$ and $\eta_{i,\mathrm{dis}}^{\mathrm{EV}}$ are the charging rate and discharging rate, respectively, and $SoC_{i,\max}^{\mathrm{EV}}$ and $SoC_{i,\min}^{\mathrm{EV}}$ represent the maximum and minimum SOC, respectively.

2.Controllable DGs

The control variables for controllable DGs are the active and reactive power output. The active and reactive power outputs of DG at node i are denoted by $P_{i,t}^{\mathrm{DG}}$ and $Q_{i,t}^{\mathrm{DG}}$, respectively. The constraints of active and reactive power are:(4)$$0\leq P_{i,t}^{\mathrm{DG}}\leq P_{i,\max}^{\mathrm{DG}},i\in \Omega_{n},t\in \Omega_{t}$$
(5)$$0\leq Q_{i,t}^{\mathrm{DG}}\leq Q_{i,\min}^{\mathrm{DG}},i\in \Omega_{n},t\in \Omega_{t}$$
where $P_{i,t}^{\mathrm{DG}}$ and $Q_{i,t}^{\mathrm{DG}}$ represent the active and reactive power of DG.

3.Third OLTCs and VRs

The OLTC/VR is controlled by changing the tap position. The control variable of the OLTC/VR on the access branch ij is expressed as $l_{ij,t}^{\mathrm{VR}}$, which varies in an integer range:(6)$$-l_{ij,\max}^{\mathrm{VR}}\leq l_{ij,t}^{\mathrm{VR}}\leq l_{ij,\max}^{\mathrm{VR}},ij\in \Omega_{b},t\in \Omega_{t}$$
where $l_{ij,\max}^{\mathrm{OLTC}}$ represents the maximum adjustable position of the OLTC/VR.

4.SCBs

The SCB adjusts the amount of reactive power it provides by regulating the number of operating units. The number of operating units of the SCB at node i is expressed as $n_{i,t}^{\mathrm{SCB}}$, which is taken in an integer range:(7)$$0\leq n_{i,t}^{\mathrm{SCB}}\leq n_{i,\max}^{\mathrm{SCB}},i\in \Omega_{n},t\in \Omega_{t}$$
where $n_{i,\max}^{\mathrm{SCB}}$ is the maximum number of units that can be connected to operation.

#### 2.1.2. Operational Constraints of the DN

The operational constraints of the distribution network are as follows:(8)$${\left(P_{t}^{S}\right)}^{2}+{\left(Q_{t}^{S}\right)}^{2}\leq {\left(S^{S}\right)}^{2},t\in \Omega_{t}$$
(9)$$V_{i,\min}\leq V_{i,t}\leq V_{i,\max},i\in \Omega_{n},t\in \Omega_{t}$$
(10)$$I_{ij,\min}\leq I_{ij,t}\leq I_{ij,\max},ij\in \Omega_{b},t\in \Omega_{t}$$

Equation (8) constrains the complex power that the substation can withstand, with $P_{t}^{S}$, $Q_{t}^{S}$, and $S^{S}$ in the equation being the active power, reactive power, and maximum apparent power of the substation, respectively. Equations (9) and (10) constrain the node voltage $V_{i,t}$ and branch current $I_{i,t}$, respectively.

#### 2.1.3. MDP Model

The challenge of modeling the collaborative problem of EV charging scheduling and voltage control in the distribution network is how to deal with various uncertainties in the system. It is also necessary to balance the charging target of EV users and the voltage control target of distribution network. Therefore, we establish an MDP model for the collaborative optimization problem of the EV charging scheduling and distribution network voltage control. The state variable, action variable, and reward function of the system are reasonably designed in the model.

1.State

The state variable of the system at any time t is defined as:(11)$$s_{t}=(P_{1,t-T+1}^{D},\cdots ,P_{1,t}^{D},Q_{1,t-T+1}^{D},\cdots ,Q_{1,t}^{D},P_{1,t}^{\mathrm{PV}},P_{1,t}^{\mathrm{WT}},SoC_{1,t}^{\mathrm{EV}},t_{1,\mathrm{arr}}^{\mathrm{EV}},t_{1,\mathrm{dep}}^{\mathrm{EV}},SoC_{1,\mathrm{dep}}^{\mathrm{EV}},\cdots ,\;P_{i,t-T+1}^{D},\cdots ,P_{i,t}^{D},Q_{i,t-T+1}^{D},\cdots ,Q_{i,t}^{D},P_{i,t}^{\mathrm{PV}},P_{i,t}^{\mathrm{WT}},SoC_{i,t}^{\mathrm{EV}},t_{i,\mathrm{arr}}^{\mathrm{EV}},t_{i,\mathrm{dep}}^{\mathrm{EV}},SoC_{i,\mathrm{dep}}^{\mathrm{EV}}\cdots ,\;R_{t-T+1}^{S},\cdots ,R_{t}^{S},t),i\in \Omega_{n},t\in \Omega_{t}$$
where $P_{i,t}^{D}$ and $Q_{i,t}^{D}$ represents the active and reactive power demand of node i, respectively, and the subscripts from $\left(t-T+1\right)$ to t−T+1 indicate the information for the past T time periods; $R_{t-T+1}^{S},\cdots ,R_{t}^{S}$ represents the historical electricity price of the past T slots; $P_{i,t}^{\mathrm{WT}}$ and $P_{i,t}^{\mathrm{PV}}$ represent the power output of wind turbine (WT) and PV power output of node i, respectively; and for the EV connected to node i, $SoC_{i,t}^{\mathrm{EV}}$ represents the its state of charge at timeslot t, $t_{i,\mathrm{arr}}^{\mathrm{EV}}$, and $t_{i,\mathrm{dep}}^{\mathrm{EV}}$ represent the start charging time and departure time, respectively, and $SoC_{i,\mathrm{dep}}^{\mathrm{EV}}$ represents the expected state of charge of EV at departure time $t_{i,\mathrm{dep}}^{\mathrm{EV}}$.

The dimension of state space will explode when the system is large. In the distribution network studied, we assume that the WT and PV are uncontrollable power supplies. To reduce the complexity of the state space, the active power demand of node i can be replaced by the net load demand of the node:(12)$$P_{i,t}^{\mathrm{Net}}=P_{i,t}^{D}-P_{i,t}^{\mathrm{PV}}-P_{i,t}^{\mathrm{WT}}$$

The simplified system state variable is represented as follows:(13)$$s_{t}=(P_{1,t-T+1}^{\mathrm{Net}},\cdots ,P_{1,t}^{\mathrm{Net}},Q_{1,t-T+1}^{D},\cdots ,Q_{1,t}^{D},SoC_{1,t}^{\mathrm{EV}},t_{1,\mathrm{arr}}^{\mathrm{EV}},t_{1,\mathrm{dep}}^{\mathrm{EV}},SoC_{1,\mathrm{dep}}^{\mathrm{EV}},\cdots ,\;P_{i,t-T+1}^{\mathrm{Net}},\cdots ,P_{i,t}^{\mathrm{Net}},Q_{i,t-T+1}^{D},\cdots ,Q_{i,t}^{D},SoC_{i,t}^{\mathrm{EV}},t_{i,\mathrm{arr}}^{\mathrm{EV}},t_{i,\mathrm{dep}}^{\mathrm{EV}},SoC_{i,\mathrm{dep}}^{\mathrm{EV}},\cdots ,\;R_{t-T+1}^{S},\cdots ,R_{t}^{S},t),i\in \Omega_{n},t\in \Omega_{t}$$

2.Action

The control variables of the system include the active and reactive power of the DG, the charging capacity of the electric vehicle, the tap position of the OLTC, and the number of SCB units.
(14)$$a_{t}=\left(P_{1,t}^{\mathrm{DG}},Q_{1,t}^{\mathrm{DG}},P_{1,t}^{\mathrm{EV}},n_{1,t}^{\mathrm{SCB}},l_{1,t}^{\mathrm{OLTC}},\cdots ,P_{i,t}^{\mathrm{DG}},Q_{i,t}^{\mathrm{DG}},P_{i,t}^{\mathrm{EV}},n_{i,t}^{\mathrm{SCB}},l_{i,t}^{\mathrm{OLTC}},\cdots \right),i\in \Omega_{n},t\in \Omega_{t}$$
where the tap position of the OLTC and the number of SCB units are discrete actions and the rest are continuous actions.

3.Reward function

The design of the reward function takes into consideration the operation cost of the system and the voltage violations of the distribution network. The operating costs of the system include the cost of the DN interacting with the main grid through the substation, the generation costs of DG and the charging cost of EVs.
(15)$$r_{t}=-\left[\alpha \left(C_{t}^{G}+\underset{i\in \Omega_{n}}{\sum }C_{i,t}^{\mathrm{DG}}+\underset{i\in \Omega_{n}}{\sum }C_{i,t}^{\mathrm{EV}}\right)+\left(1-\alpha \right)\underset{i\in \Omega_{n}}{\sum }C_{i,t}^{V}\right],t\in \Omega_{t}$$
where α represents the weight coefficient, which indicates that the reward function is the tradeoff value between the operating cost and the voltage stability target.

The first term $C_{t}^{G}$ calculates the cost of purchasing electricity from a power exchange station.
(16)$$C_{t}^{G}=P_{t}^{S}R_{t}^{S}\Delta t$$

The second term $C_{i,t}^{\mathrm{DG}}$ calculates the generation cost of the DG at node i.
(17)$$C_{i,t}^{\mathrm{DG}}=a_{i}{\left(P_{i,t}^{\mathrm{DG}}\right)}^{2}+b_{i}P_{i,t}^{\mathrm{DG}}+c_{i}$$
where $a_{i}$, $b_{i}$, and $c_{i}$ are the generation cost coefficients.

The third term $C_{i,t}^{\mathrm{EV}}$ calculates the charging cost of EVs.
(18)$$C_{i,t}^{\mathrm{EV}}=P_{i,t}^{\mathrm{EV}}R_{i,t}^{S}\Delta t$$

We assume that all EVs in the DN system are connected through V2G to take full advantage of the flexibility of EVs to participate in regulating the voltage of the DN, and that the tariff is settled according to the trading tariff of the substation.

The fourth term $C_{i,t}^{V}$ calculates the penalty of voltage violation, corresponding to the node voltage constraint given in Equation (9).
(19)$$C_{i,t}^{V}=\sigma \cdot \left(\max\left(0,\;V_{i,t}-V_{i,\max}\right)+\max\left(0,\;V_{i,\min}-V_{i,t}\right)\right)$$
where σ is the penalty coefficient of voltage deviation.

4.Objective

Define $J\left(\pi \right)$ as the expected cumulative discount return over the scheduling cycle:(20)$$J\left(\pi \right)=E_{\tau ∼\pi }\left[r_{0}+\gamma r_{1}+\cdots +\gamma^{T-1}r_{T}\right]$$
where $\gamma \in \left[0,1\right]$ is the discount factor and τ represents the system trajectory under policy π.

### 2.2. Deep Reinforcement Learning Solution

In this section, a DRL-based approach is introduced for collaborative EV charging scheduling and distribution network voltage control. The problem in Section 2.1 is first transformed into the RL framework. Then we design a DNN to handle the output of mixed discrete and continuous actions and train the DNN by DDPG [32]. The agent of DDPG consists of two layers. All dispatching signals and control signals are output by the upper layer of the agent, and the lower layer obtains a complete reward value by controlling the controllable units in the distribution network.

#### 2.2.1. DRL Based Approach

Under the reinforcement learning framework, the learning agent interacts with the constructed MDP environment model. The optimization problem is transformed into a standard reinforcement learning framework with the following objective:(21)$$V^{\pi^{*}}\left(s_{t}\right)=\underset{a_{t}\in A\left(s_{t}\right)}{\max}Q^{\pi^{*}}\left(s_{t},a_{t}\right)$$
where $Q^{\pi^{*}}$ is the optimal action–value function. The action–value function $Q^{\pi }\left(s_{t},a_{t}\right)$ describes the expected rewards for taking action $a_{t}$ and then following policy π in state $s_{t}$, which is used in many reinforcement learning algorithms. It is denoted by Equation (22).
(22)$$Q^{\pi }\left(s_{t},a_{t}\right)=E_{r_{i>t},s_{i>t},a_{i>t}\sim \pi }\left[r_{t}\left|s_{t},a_{t}\right.\right]$$

The optimal action–value function $Q^{\pi^{*}}$ can be derived by solving the Bellman equation recursively. Then, we can obtain the optimal policy $\pi^{*}$, which means the optimal action $a_{t}\sim \pi^{*}$ can be obtained. This optimization problem can be described by Equations (23) and (24).
(23)$$Q^{\pi^{*}}\left(s_{t},a_{t}\right)=E_{\pi^{*}}\left[r_{t}+\gamma \cdot \underset{a_{t+1}\in A\left(s_{t}\right)}{\max}Q^{\pi^{*}}\left(s_{t+1},a_{t+1}\right)\right]$$
(24)$$\pi^{*}\left(s_{t}\right)=\arg\underset{a_{t}\in A\left(s_{t}\right)}{\max}Q^{\pi^{*}}\left(s_{t},a_{t}\right)$$

The Bellman equation will be difficult to solve when faced with complex problems. To address this problem, value-based methods use look-up table or deep neural network to estimate the optimal action–value function $Q^{\pi^{*}}$ and update it iteratively. The approximation function is usually described in the form of a function $Q\left(s,a|\theta^{Q}\right)$ with respect to the parameters $\theta^{Q}$ and the parameters are optimized with the objective of minimizing the loss function Loss based on the temporal difference theory.
(25)$$Loss\left(\theta^{Q}\right)=E_{s_{t}\sim \rho^{\beta },a_{t}\sim \beta }\left[{\left(y_{t}-Q\left(s_{t},a_{t}|\theta^{Q}\right)\right)}^{2}\right]$$
where B is the batch size of the samples sampled from the replay buffer and $y_{t}$ is the target value:(26)$$y_{t}=r_{t}\left(s_{t},a_{t}\right)+\gamma \cdot Q\left(s_{t+1},\mu \left(s_{t+1}\right)|\theta^{Q}\right)$$

Reinforcement learning that uses an approximation function to estimate the value function is known as value-based RL methods. However, they have some disadvantages in practical applications, especially when dealing with problems with continuous action spaces where a good scheduling strategy cannot be obtained. Therefore, we use policy-based reinforcement learning methods, which directly approximate the policy and optimize the policy function through the gradient ascent method until a convergent policy is obtained.

The deep deterministic policy gradient (DDPG) [32] algorithm is introduced to solve the complex coordinate EV charging and voltage control problem with high-dimensional and continuous action spaces by only using low-dimensional observations. The DDPG algorithm is a policy-based DRL algorithm with actor–critic architecture. Both actor and critic contain two neural networks, with actor consisting of two DNN with parameters $\theta^{\mu }$ and $\theta^{\mu \prime }$, and critic consisting of two multilayer perceptron (MLP) with parameters $\theta^{Q}$ and $\theta^{Q\prime }$, respectively. The construction of the DDPG algorithm is shown in Figure 2. Similar to standard reinforcement learning, DDPG has a learning agent that interacts with a distribution network environment in discrete timesteps. The input of the DDPG agent is the system state $s_{t}$ at time step t and the output is action $a_{t}$. We assume the studied DN environment is fully observed. To ensure independence between samples when using neural networks, DDPG uses experience replay technology to ensure independence between the samples used for target value updating. After each interaction of the agent with the environment, we can obtain a sample containing $s_{t}$, $a_{t}$, $r_{t}$, and $s_{t+1}$, and store this sample in the replay buffer. The agent continues to interact with the environment until the set condition is met, then B samples are randomly sampled from the replay buffer to minimize the loss (Equation (25)) of the critic network and to calculate the gradient (Equation (27)) of the actor network to softly update the parameters of the critic and actor networks, respectively.

The DDPG algorithm combines the success of the actor-critic approach and DQN [33] using dual networks on top of the deterministic policy gradient (DPG) algorithm. The DPG algorithm is based on the actor–critic structure, which consists of an actor and a critic. The critic $Q\left(s,a\right)$ is learned using the Bellman equation as in Q-learning. According to Equation (26), the update rule for the parameters of the critic is given by Equation (27).
(27)$$Loss\left(\theta^{Q}\right)=\frac{1}{B}\sum_{i}{\left(y_{i}-Q\left(s_{i},a_{i}|\theta^{Q}\right)\right)}^{2}$$

The actor is a parameterized actor function $\mu \left(s|\theta^{\mu }\right)$ that specifies the current policy by deterministically mapping states to actions. The parameters of the actor’s value network are updated based on the policy gradient method. The policy gradient algorithms apply a gradient ascent method to update policy parameters and rely on the sampled sequence of decisions when interacting with the environment. The actor is updated by following the applying the chain rule to the expected return from the start distribution J. The update rule for the parameters of the actor is given by Equation (28):(28)$$\nabla_{\theta^{\mu }}J\approx E_{s_{t}\sim \rho^{\beta }}\left[\nabla_{\theta^{\mu }}Q\left(s,a|\theta^{Q}\right){|}_{s=s_{t},a=\mu \left(s_{t}|\theta^{\mu }\right)}\right]\;\;\;\;\;\;\;\;\;\;=E_{s_{t}\sim \rho^{\beta }}\left[\nabla_{a}Q\left(s,a|\theta^{Q}\right){|}_{s=s_{t},a=\mu \left(s_{t}\right)}\nabla_{\theta_{\mu }}\mu \left(s|\theta^{\mu }\right){|}_{s=s_{t}}\right]\;\;\;\;\;\;\;\;\;\;\;\approx \frac{1}{B}\underset{i}{\sum }\nabla_{a}Q\left(s,a\left|\theta^{Q}\right.\right){|}_{s=s_{i},a=\mu \left(s_{i}\right)}\cdot \nabla_{\theta^{\mu }}\mu \left(s|\theta^{\mu }\right){|}_{s_{i}}$$
where J is the expected return from the start distribution, μ is the deterministic target policy, θ is the parameter of the function approximator, ρ is the discounted state visitation distribution for policy, β is a different stochastic behavior policy, and $s_{i}$ is the state of the ith sample in the small batch of samples sampled from the replay buffer.

#### 2.2.2. Design of the Parameterized Policy Network

The proposed DDPG uses a multilayer perceptron (MLP) to approximate the policy and output the continuous action value. We design a DNN to approximate the coordinated policy. Figure 3 illustrates the architecture of the designed policy network. The status information on the system’s renewable energy output $P_{t}^{\mathrm{WT}},P_{t}^{\mathrm{PV}}$, load demand $P_{t}^{L}$, real-time LMP price $R_{t}^{S}$, and SOC of EV $SoC_{t}$ is fed into the network and output as a defined continuous action vector. To ensure the stability and convergence of the learning process, all input state data are normalized according to their respective min–max values. RNN can be used as a policy network when the state variables contain information from the past T time periods. In our model, the state variables only contain information from the current moment to reduce the dimensionality of the state space, so we choose a DNN as the policy network to extract the feature information of the system state variables. The final layer of the network uses tanh as the activation function and outputs continuous values in the range [−1, 1]. To output discrete control actions, we add an integral block behind the output layer to output discrete control signals to OLTC and CB. In this way, the mixed discrete continuous action output $a_{t}$ is obtained. The min–max block in the figure behind the output layer represents the limit on the range of output continuous actions, corresponding to the constraints on DGs and EV in Section 2. To alleviate the problem of a vanishing gradient or exploding gradient, a rectified linear unit (ReLU) is used as the activation function of each neuron in the hidden layer. The details of the architecture of the policy network of the proposed DDPG structure is provided in the Table 1.

#### 2.2.3. Practices Implementation

The scheduling process for DN can be summarized as the offline training and online scheduling process in Figure 4. The coordinate EV charging and voltage control strategy in the agent contains two layers. The upper layer is the dispatching layer, which outputs the control signals of all dispatchable units according to the system status. The lower layer is the voltage control layer, which is the response for receiving these control signals and controlling the dispatchable units in the DN system. The parameters (weights and biases) of the initial policy of the agent are random and the policy network cannot output the optimal action. Therefore, the policy network of the agent needs to be trained offline using historical environmental data before it can operate practically. The parameters of the DNN are updated through iterative interaction with the environment and the accumulation of experience. With this approach, the agent can gradually optimize the network parameters to more accurately approach the optimal collaborative strategy.

The agent is trained in a centralized mode using historical system data and then run in online mode. During the training process, the voltage layer calculates the penalty of voltage fluctuation in the reward function by running a simulated distribution network system. The pseudocode for the training procedure of the DRL-based method is presented in Algorithm 1.
**Algorithm 1** DDPG-based Learning Algorithm1**Initialize** weights $\theta^{Q}$ and $\theta^{\mu }$ of critic network $Q\left(s,a|\theta^{Q}\right)$ and actor network $\mu \left(s|\theta^{\mu }\right)$
2**Initialize** weights $\theta^{Q\prime }\leftarrow \theta^{Q}$, $\theta^{\mu \prime }\leftarrow \theta^{\mu }$ of target network Q′ and μ′
3**Initialize** experience replay buffer R
4**for** episode = 1, 2, …, M **do**5 Receive initial observation state $s_{1}$
6 **for** t = 1, 2, …, T **do**7  Choose $a_{t}=\mu \left(s_{t}|\theta^{\mu }\right)$ and do simulation using pandapower8  Observe reward $r_{t}$ and the next state $s_{t+1}$
9  Store transition $\left(s_{t},a_{t},r_{1},s_{t+1}\right)$ in R
10  Sample a random minibatch of B transitions $\left(s_{i},a_{i},r_{i},s_{i+1}\right)$ from R
11  Set $y_{i}=r_{i}+\gamma \cdot Q\prime \left(s_{i+1},\mu \prime \left(s_{i+1}|\theta^{\mu \prime }\right)|\theta^{Q\prime }\right)$ according to Equation (26)12  Update critic network parameters by minimizing the loss, see Equation (27):      $Loss=\frac{1}{B}\sum_{i}{\left(y_{i}-Q\left(s_{i},a_{i}|\theta^{Q}\right)\right)}^{2}$
13  Update the actor policy using the sampled policy gradient, see Equation (28):      $\nabla_{\theta^{\mu }}J\approx \frac{1}{B}\underset{i}{\sum }\nabla_{a}Q\left(s,a\left|\theta^{Q}\right.\right){|}_{s=s_{i},a=\mu \left(s_{i}\right)}\cdot \nabla_{\theta^{\mu }}\mu \left(s|\theta^{\mu }\right){|}_{s_{i}}$
14  Softly update the target networks using the updated critic and actor network parameters:      $\theta^{Q\prime }\leftarrow \tau \theta^{Q}+\left(1-\tau \right)\theta^{Q}$     $\theta^{\mu \prime }\leftarrow \tau \theta^{\mu }+\left(1-\tau \right)\theta^{\mu }$
15 **end for**16**end for**

In Algorithm 1, all network parameters (weights and bias) of the DDPG are initialized before starting training. At the beginning of each episode, the environment is reset in order to obtain the initial state of the system. Then, the policy network under the current parameters is used to interact with the environment for T time steps. During the interaction, the immediate reward, the observed state at the next moment, current state and the action are composed to be one sample, and this sample is stored in the replay buffer. Next, a random batch of samples from the replay buffer is used to update the parameters of the actor and critic networks of DDPG according to the conditions.

After the offline training, the trained network parameters are preserved for online operation. In practical operation, the preserved network parameters are loaded, and the system state is input to output the control signals for the collaborative strategy. The agent only outputs the control signal to the system through the dispatching layer, and the voltage control layer no longer calculates the penalty of voltage fluctuation. The pseudocode for the practical running process of the algorithm is presented in Algorithm 2.
**Algorithm 2** Online Running Algorithm1**Input** system state $s_{t}$
2**Output** EV charging/discharging schedule and voltage control signals3**for** t = 1, 2, …, T **do**4 Obtain historical information and EV charging demand5 Build observation state $s_{t}$ according to Equation (13)6 Choose action $a_{t}$ according to Equation (24) using the trained Algorithm 17 Output EV charging/discharging schedule and voltage control signals8**end for**

## 3. Results and Discussion

In this section, we present the details of simulation experiments to test the proposed method and prove the effectiveness of the method through the analysis of the simulation results. The simulations are trained and tested using a personal computer with an NVIDIA RTX-3070 GPU and one Intel (R) Cores (TM) i7-10700K CPU. The code is written in Python 3.7.8, the reinforcement learning algorithm is implemented using the deep learning package TensorFlow 1.14.0 [34], and the distribution network environment is realized using pandapower 2.10.1 [35].

### 3.1. IEEE-33 Node System and Parameter Settings

The performance of the proposed learning method is evaluated on a modified IEEE-33 node system [36]. Figure 5 shows the topology of the test feeder system. An OLTC is set at bus 0 to connect to the external grid, which has 11 tap positions with an adjustment range of −10% to 10% (2% per tap). Two SCBs with a capacity of 400 kVar are connected at node 17 and node 30, each containing four units. A controllable DG is connected at node 17 and node 32, respectively, and a WT and a PV are provided at nodes 21 and 24, respectively. The detailed parameter settings of DGs and RES are presented in Table 2. To reflect the complexity of the system, we evenly distributed the EV charging stations throughout the test system. As shown in Figure 5, EV charging stations are set up on nodes 8, 13, 19, 22, and 29, each of which can be connected to a different number of EVs. Nissan Leaf is considered a typical EV prototype, and the maximum charge/discharge power for each vehicle is set at 6 kW and the battery capacity is set at 24 kWh. The charging and discharging efficiency of EVs is set at 0.98 and 0.95, respectively. As suggested by [3,37], the EV arrive time, departure time, and battery SOC at the arrival time obey truncated normal distribution. These distributions and the specific parameter settings are presented in Table 3. The safe range for the SOC of EV battery is [0.2, 1.0]. The objective is to minimize the total operating cost of the system and the fluctuation of node voltage. The safe range of nodal voltages is set between 0.95 p.u. and 1.05 p.u.

The time-series data in the California Independent System Operator (CAISO) [38] are used to simulate the electricity prices, load demand, and RES generation in the distribution network system. We downloaded data for 2019 and 2020 and used these two years as the training set and test set, respectively. To ensure the load data meet the requirements of the considered system, it is necessary to process the downloaded load data. First, normalize the downloaded load data and then multiply the node base load power of the standard IEEE-33 node system for setting. The output data of the downloaded wind turbine and photovoltaic are processed in the same way.

To verify the effectiveness and scalability of the coordinated strategy, two simulation cases are designed based on the load capacity that the system can handle: Case 1 contains 5 EVs and Case 2 contains 50 EVs. Electric vehicles have a characteristic of having more parking time than driving time, and EV users prefer to charge at night when electricity prices are lower. Therefore, we set up fewer EVs in the simulation scenario of Case 1 and more EVs in the simulation scenario of Case 2. In Case 1, the EVs considered in the system are charged during the daytime, with a charging scheduling time range of 8:00 a.m. to 19:00 p.m. for a total of 12 h. In Case 2, the EVs are charged during the nighttime, with a charging time range of 20:00 p.m. to 7:00 a.m. for a total of 12 h. The expected charge level would be no less than 80% to alleviate low range anxiety.

The proposed method is compared with several benchmark approaches, including DRL-based DQN [33], soft actor–critic (SAC) [39] and proximal policy optimization (PPO) method [40]. The policy network of DQN contains three hidden ReLU layers and 64 neurons each, and SAC has the same policy network structure as DDPG, both containing three ReLU layers with 256, 128, and 64 neurons, respectively, and an output layer with tanh as the activation function and using the same approach to obtain hybrid actions. Eleven levels of optional actions are set in the action space of DQN, and SAC and PPO output actions are present in the same way as DDPG. Additional parameters considering the algorithm are given in Table 4. These algorithms choose the same parameter to realize the voltage fluctuation to ensure the competitiveness of the comparison results.

### 3.2. Simulation Comparison of Voltage Control Performance

Figure 6 shows the results of the system nodal voltages using the DDPG algorithm. Figure 6a,c shows the results for the node voltage with voltage control in Case 1 and Case 2, respectively, and Figure 6b,d shows the results for node voltage without voltage control in Case 1 and Case 2, respectively. Comparing Figure 6a,c and Figure 6b,d vertically, it can be observed that the voltage of the system decreases as the load of the electric vehicles in the system increases. The comparison results show that controlling the voltage while scheduling the electric vehicle allows the voltage at each node in the system to be in the safe range, indicating that our proposed coordinated control strategy can implement the safe control of the node voltage of the system.

Figure 7 compares the average cumulative voltage violation (CVV) of the proposed DDPG method and the comparison method for five independent runs of the training process with different random seeds. As shown in Figure 7a, in Case 1, DDPG learns a safe and stable voltage control strategy after 500 training episodes. However, the SAC in the comparison algorithm converge after 1000 episodes of training, and the DQN and PPO converge after more than 2500 episodes. In Case 2, DDPG can converge to a lower average CVV after 1000 training episodes. Both PPO and SAC require 1500 training episodes to converge, and DQN, although converging after 2000 training episodes, still has large fluctuations, which are related to the discretization of its action output. The comparison algorithms have poorer performance in voltage control, both exhibiting higher values of voltage violations.

The control results of the coordinated control strategy for the OLTC and SCB in Case 1 and Case 2 are given in Figure 8. Combined with the node voltage results in Figure 6a,c, both the OLTC and SCB can be controlled to ensure that the voltage at each node of the system is in the range of [0.95, 1.05], thus avoiding voltage dropout and voltage overrun.

### 3.3. Simulation Comparison of Cost Reduction Performance

Figure 9 compares the cumulative operating cost curves of the proposed method with several other EV scheduling strategies, including the constant power charging strategy (CPC), TOU excitation strategy and PSO-based scheduling strategy without voltage control (NVC). In the constant power charging strategy, the charging power of the EV is set to the maximum charging power of the battery. The TOU price used in the simulation is given in Table 5 [20]. As shown in Figure 9a, the cumulative operating cost for DDPG, CPC, TOU, and NVC in Case 1 are USD 0.974M, USD 1.219M, USD 1.096M, and USD 0.722M, respectively. Compared with CPC and TOU, DDPG reduces the operating cost by 20.1% and 11.1%. As shown in Figure 9b, the cumulative operating cost for DDPG, CPC, TOU, and NVC in Case 2 are USD 9.35M, USD 13.468M, USD 10.272M, and USD 7.309M, respectively. Compared with CPC and TOU, DDPG reduces the operating cost by 30.58% and 8.98%. Although the scheduling strategy without voltage control has the lowest cumulative operating cost, it cannot guarantee the stability of the system voltage. Combined with the above analysis, the following conclusion can be drawn. Our proposed DDPG approach ensures system voltage stability at the expense of some economy.

Figure 10 compares the cumulative rewards of the proposed DDPG method and the comparison method for five independent runs of the training process with different random seeds. Figure 10a shows the convergence curves of the different DRL methods in Case 1. From Figure 10a it can be seen that DDPG is able to learn an economical EV charging strategy after 500 episodes of training. The SAC in the comparison method is measured to converge after 1500 episodes, while the DQN requires more training to converge. The cumulative return of DDPG, DQN, SAC, and PPO during the training process converge to USD 1.2135M, USD 2.5023M, USD 1.8798M, and USD 1.4392M, respectively. Compared to DQN, SAC, and PPO, DDPG can reduce the cost by 51.5%, 35.4%, and 15.7%, respectively. Figure 10b shows the convergence curves of the different DRL methods in Case 2. DDPG in Figure 10b converges after 1000 training episodes, while PPO, SAC, and DQN all requires 2000 training episodes to converge. The cumulative return of DDPG, DQN, SAC, and PPO are USD 3.2181M, USD 4.6912M, USD 4.4152M, and USD 3.7642M, respectively. Compared to DQN, SAC, and PPO, DDPG can reduce the cost by 31.4%, 27.11%, and 14.51%, respectively. Additionally, as can be seen in Figure 10, DDPG and PPO are able to achieve higher returns than SAC and DQN. Compared to PPO, DDPG has a faster convergence rate and more stable convergence results. Combined with the training results in Figure 7, we can conclude that DDPG has a faster and more stable performance than popular DRL methods in learning a safe and economical coordinated control strategy. The average running time of training and testing (one-step) of all algorithms is listed in Table 6. DQN has the longest training time compared to the DRL method that outputs continuous actions because it has a discrete action space. As the number of controllable units in the system increases, the training time of DQN increases as the action space increases.

The scheduling results for DGs and EVs in Case 1 are given in Figure 11. The purple star symbol on the SOC curve in Figure 11b indicates the SOC value at the start of EV charging. From the SOC curve the following conclusions can be drawn, our proposed charging strategy can charge the EV with goal of reducing charging costs and the reward function is designed to balance the EV charging target with the DN voltage control target. Based on the SOC values at the end of charging process, the battery of the EV is not always fully charged; this is because we consider the voltage stability of the DN when scheduling the EV for charging. The voltage constraint of the DN prevents the EVs from being perfectly filled but the desired level can still be achieved.

## 4. Conclusions

In this paper, we proposed a DRL approach based on DDPG for coordinate EV charging and voltage control problems in distribution networks. The proposed two-layer scheduling control strategy enables the agent to learn an economical scheduling strategy and maintain the voltage stability of the distribution network. The proposed method is data-driven and does not rely on uncertain models in the system. The designed policy network can directly generate hybrid continuous scheduling and discrete control signals. The simulation experiment is tested on a modified IEEE-33 node system and the real-world power system data are used for training and testing. Two simulation cases of different scenarios are designed to verify the effectiveness and scalability of the proposed approach. Simulation results demonstrate that the proposed approach can successfully learn an effective policy to charge EVs in a cost-efficient way, considering voltage stability. The numerical results demonstrate the effectiveness of the DDPG approach, which can significantly reduce the operating cost of the system in both Case 1 and in Case 2 scenarios and has a faster convergence rate compared to the other DRL methods used for comparison. The comparison results show that the proposed approach is well balanced to take into account the charging demand of EVs and the voltage stability of the distribution network.

The charging scheduling of EVs is a complex process, and more physical characteristics should be considered. For future work, the impacts of battery degradation and V2G operation on the EV charging process should be carefully considered in order to establish a more realistic environment.

## Figures and Tables

**Figure 1 sensors-23-01618-f001:**
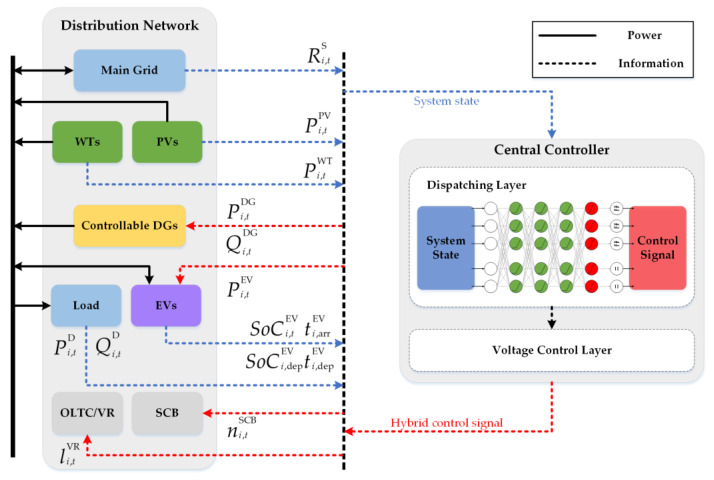
The collaborative EV charging and voltage control framework.

**Figure 2 sensors-23-01618-f002:**
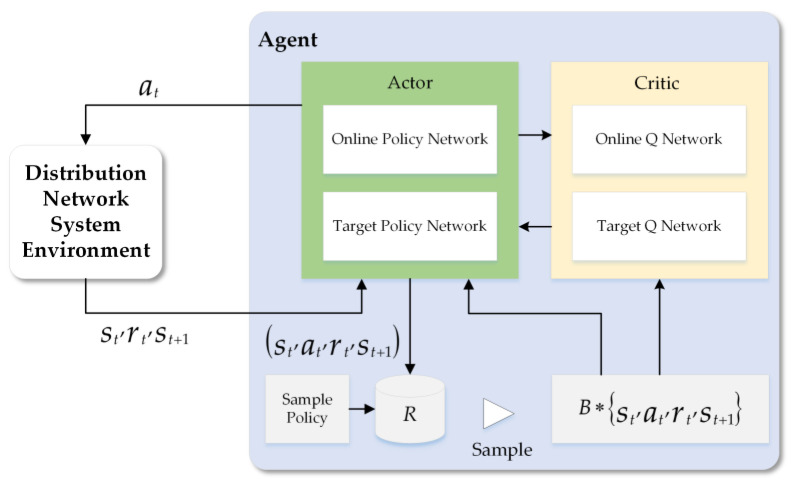
Constructure of the proposed DDPG algorithm.

**Figure 3 sensors-23-01618-f003:**
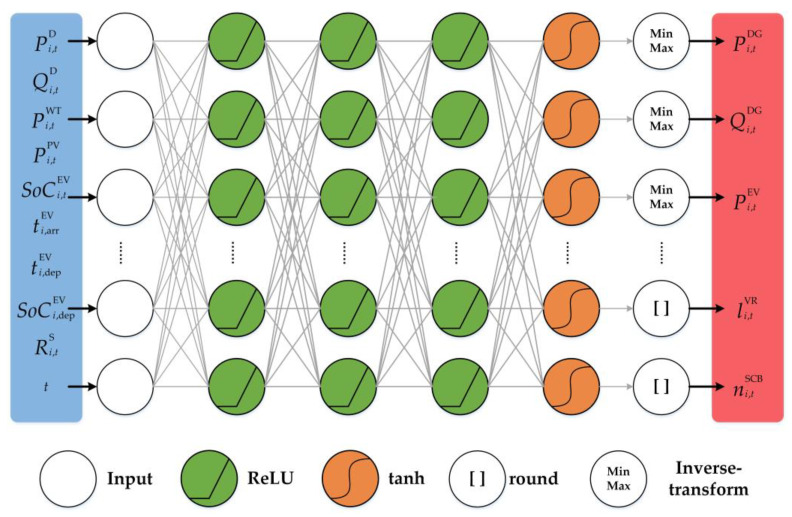
The architecture of the designed policy network.

**Figure 4 sensors-23-01618-f004:**
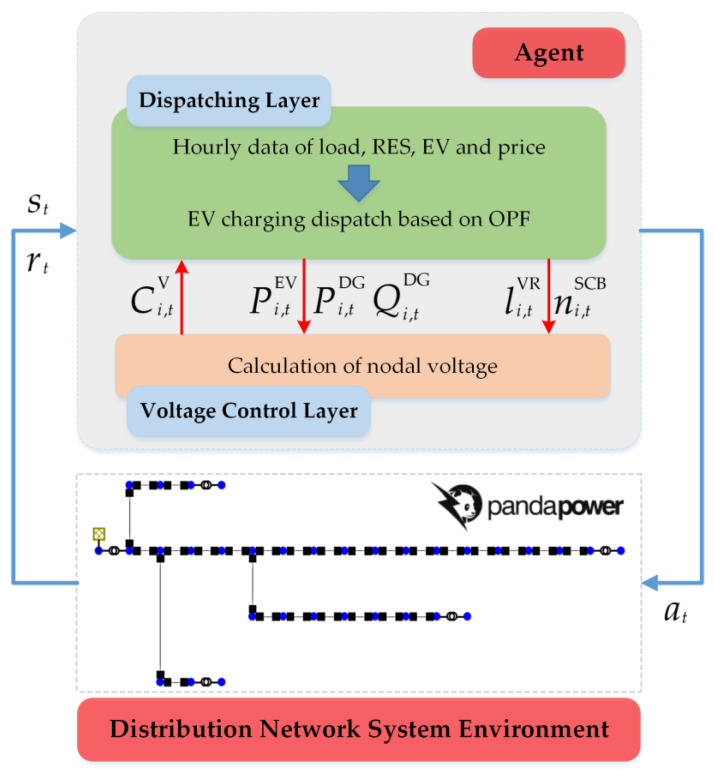
Schematic diagram of the framework for offline training and online operation.

**Figure 5 sensors-23-01618-f005:**
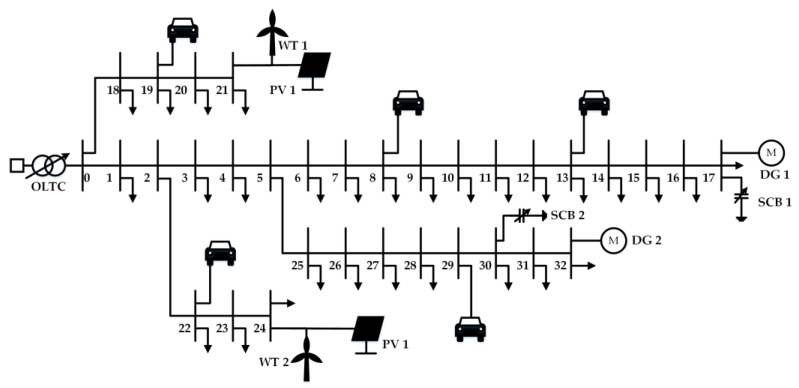
Modified IEEE 33-bus distribution network. The number on the feeder (0–32) indicates the number of the node.

**Figure 6 sensors-23-01618-f006:**
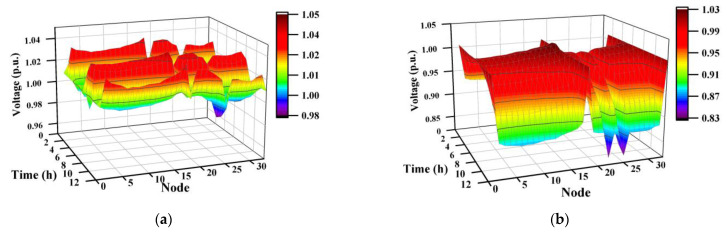
Comparison of node voltage between the coordinated solution and dispatch-only solution. (**a**) Coordinated solution of Case 1; (**b**) dispatch-only solution of Case 1; (**c**) coordinated solution of Case 2; (**d**) dispatch-only solution of Case 2.

**Figure 7 sensors-23-01618-f007:**
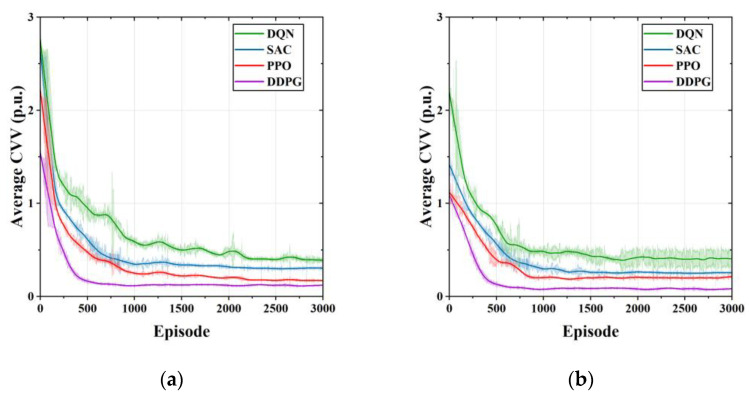
Comparison of average cumulative voltage violation during the training process for different learning algorithms. (**a**) Average CVV of Case 1; (**b**) average CVV of Case 2.

**Figure 8 sensors-23-01618-f008:**
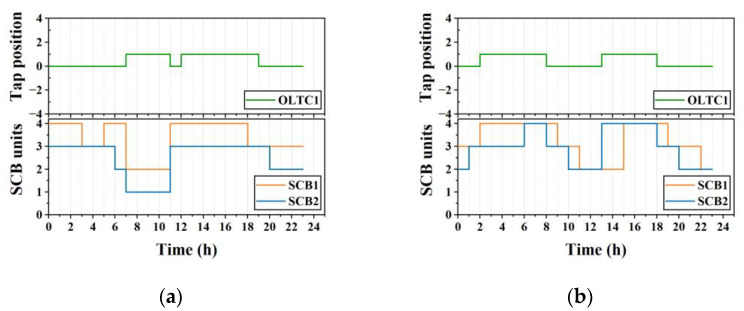
Control results for the OLTC and SCB. (**a**) Results of Case 1; (**b**) results of Case 2.

**Figure 9 sensors-23-01618-f009:**
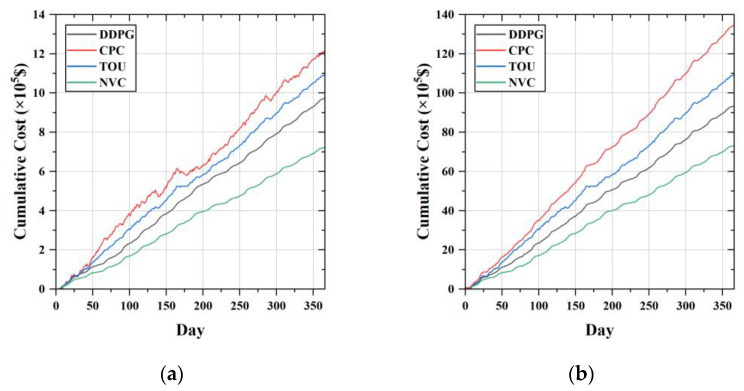
Comparison of the cumulative cost of different scheduling methods on the test set. (**a**) Comparison results of Case 1; (**b**) comparison results of Case 2.

**Figure 10 sensors-23-01618-f010:**
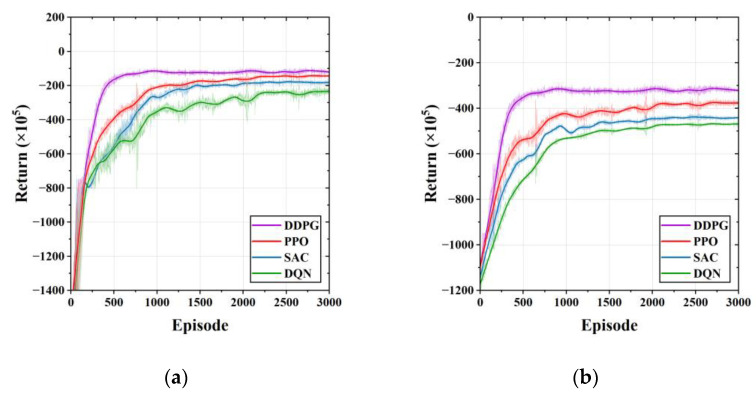
Comparison of average reward during the training process for different learning algorithms. (**a**) Comparison results of Case 1; (**b**) comparison results of Case 2.

**Figure 11 sensors-23-01618-f011:**
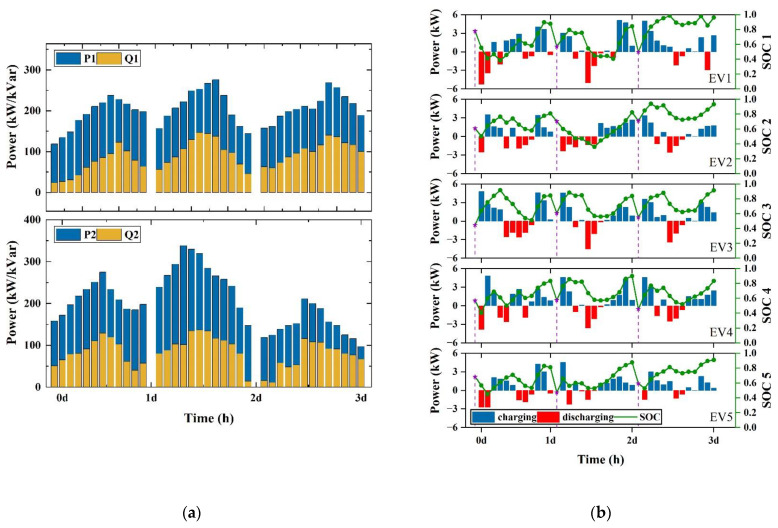
Operation results of DDPG for the IEEE-33 node system. (**a**) Active and reactive power generation of DG1 and DG2; (**b**) charging/discharging power and energy status of EVs.

**Table 1 sensors-23-01618-t001:** Policy network structure.

Layer	Output Dimension
Input layer (state space)	$N^{S}$
Full connection layer + ReLU (units 256)	256
Full connection layer + ReLU (units 128)	128
Full connection layer + ReLU (units 64)	64
Full connection layer + tanh (action dimension)	$N^{A}$
Round block and inverse-transform block	$N^{A}=N_{D}^{A}+N_{C}^{A}$
Output of hybrid action = $N^{A}$	

**Table 2 sensors-23-01618-t002:** Operation parameters of controllable units in the distribution network.

Type and Number		Parameters					
DG	NO.	Maximum Power (kW)	Minimum Power (kW)	a (USD/kWh^2^)		b (USD/kWh)	c (USD/h)
	1	300	100	0.0175		1.75	0
	2	400	100	0.0625		1	0
RES	NO.	Maximum power (kW)			Minimum power (kW)		
	WT1-2	15			0		
	PV1-2	8			0		

**Table 3 sensors-23-01618-t003:** Parameter setting of EVs.

Variable	Distribution	Boundary
Arrival time	$t_{\mathrm{arr}}^{\mathrm{EV}}\sim N\left(9,\;1^{2}\right)$	$8\leq t_{\mathrm{arr}}^{\mathrm{EV}}\leq 10$
Departure time	$t_{\mathrm{dep}}^{\mathrm{EV}}\sim N\left(18,\;1^{2}\right)$	$17\leq t_{\mathrm{dep}}^{\mathrm{EV}}\leq 19$
Initial SOC	$SoC_{\mathrm{dep}}^{\mathrm{EV}}\sim N\left(0.6,\;0.1^{2}\right)$	$0.4\leq SoC_{\mathrm{dep}}^{\mathrm{EV}}\leq 0.8$

**Table 4 sensors-23-01618-t004:** Parameter setting of the algorithm.

Symbol	Parameters	Numerical
M	Training episode	3000
$lr^{a}$	Learning rate of actor	0.00001
$lr^{c}$	Learning rate of critic	0.001
τ	Soft update coefficient	0.01
R	Memory capacity	25,000
B	Batch size	48
γ	Discount factor	0.95
α	Trade-off factor	0.5
σ	Penalty of voltage fluctuation	100,000

**Table 5 sensors-23-01618-t005:** Time of use electric price.

Type	Time Period	Price (USD/kWh)
Valley	1:00–8:00	0.295
Peak	9:00–12:00, 18:00–21:00	0.845
Flat	13:00–17:00, 22:00–24:00	0.56

**Table 6 sensors-23-01618-t006:** Average time consumption on training and online computation by different learning algorithms.

		DDPG	DQN	SAC	PPO
Case 1	Training (h)	13.57	28.36	18.64	16.85
	Testing (s)	0.0014	0.0016	0.0014	0.0015
Case 2	Training (h)	14.85	40.46	20.81	18.72
	Testing (s)	0.0024	0.0032	0.0026	0.0027

## Data Availability

Not applicable.

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
