# Peer review of "Deep Reinforcement Learning for Charging Scheduling of Electric Vehicles Considering Distribution Network Voltage Stability"

_sensors, 2023, doi:10.3390/s23031618_

Round 1

Reviewer 1 Report

No big gaps in article, complex when compare with other studies by this team. Pictures are small in my eyes.

Reviewer 2 Report

The paper is written; clearly, this is a timely subject.

 There are some concerns about the paper’s contributions and uncertainty modelling. Detailed comments are given as follows:

1-     There are some concerns about the introduced contributions; for instance, considering a two-level control strategy doesn’t seem like a novel solution in the presented form. Please add more description regarding the mentioned contributions.

2-     As one of the primary duties of the proposed DRL method is to cover the uncertainty of the EV load demand, the literature survey part should be enriched with more studies that focus on the modelling of EV load demand in the context of the energy market, following papers can be considered as suitable sources for this approach: A Novel Cross-Case Electric Vehicle Demand Modeling Based on 3D Convolutional Generative Adversarial Networks, Plug-in electric vehicle behaviour modelling in the energy market: A novel deep learning-based approach with clustering technique. These approaches try to estimate the day-ahead demand of the users by applying advanced AI techniques, including 3D convolutional networks and LSTM networks and then use deterministic optimization solutions for handling the optimal charging task. Voltage stability can also be defined as one of the objectives of the optimization process. Please explain the main benefits of employing DRL methods against the aforementioned approaches.

3-     Please add more scenarios in the numerical results, for instance, considering a different number of EVs and the impacts of the electricity price in the simulation results.

4-     Please add more discussion about Fig. 3. More details about the network are needed to verify the overall idea of this study.

5-     The conclusion part should be updated with more details about the general outcome of this study.

6-     Please explain the impacts of the battery degradation and V2G operation on the proposed method. If possible, add them to your simulation results; if not, explain them as future works for this study.

Reviewer 3 Report

1. The research gap should be mentioned through critical reviews. Some relevant references suggested are-

Charging cost minimisation by centralised controlled charging of electric vehicles-International Transactions on Electrical Energy Systems

Grid integration of electric vehicles for economic benefits: A review-Journal of Modern Power Systems and Clean Energy

Grid integration of battery swapping station: A review-Journal of Energy Storage

Economic Operation Scheduling of Microgrid Integrated with Battery Swapping Station-Arabian Journal for Science and Engineering   

2.  Improve quality of diagrams. 

Reviewer 4 Report

This paper is dealing with Deep reinforcement learning for charging scheduling of electric vehicles considering distribution network voltage stability. The topic is interesting, however, there are some comments before more consideration in this journal.

1- The Deep Reinforcement Learning section should be extended and more details regarding the methodology should be added.

2- Term Update the actor policy using the sampled policy gradient in Algorithm 1 needs more discussion. How it is calculated? Please refer to the library/function developed for this specific section. 

3- Equation 2 is not true since the charging and discharging efficiencies are not the same, especially for EVs. Please refer to: A two-stage joint operation and planning model for sizing and siting of electrical energy storage devices considering demand response programs

4- It is very important to provide the data for EVs in terms of size, aggregation and other specific data for them used in the simulations.

5- Why do the Authors provide the minimum for RES units, as they are not dispatchable? For example, the PV during nighttime couldn't provide energy. 

6- Figure 6. Comparison of node voltage between the coordinated solution and dispatch-only solution. (a) Coordinated solution; (b) Dispatch-only solution.  Why do the Authors depict only 12 hours of the day?

7- Is there any defence on 'The EVs considered in the system are charged only during the daytime, with a charging scheduling time range of 8:00 a.m. to 19:00 p.m. for a total of 12 hours. And the expected charge level is not less than 80%.'. It is much preferred to charge the EVs during the rest time at home, i.e., during the nighttime with minimum tariffs.

8- Please provide the reference for the Tariff. 

9- Providing the data/model/code in GitHub is fully recommended. 

Round 2

Reviewer 2 Report

The paper is ready for publication.

Reviewer 4 Report

The Authors covered the main concerns of this Reviewer and the manuscript can ve accepted as it is.